# Explosive neural networks via higher-order interactions in curved statistical manifolds

Miguel Aguilera [1,2] ✉, Pablo A. Morales[3,4], Fernando E. Rosas [5,6,7,8] & Hideaki Shimazaki [9,10]

Higher-order interactions underlie complex phenomena in systems such as biological and artificial neural networks, but their study is challenging due to the scarcity of tractable models. By leveraging a generalisation of the maximum entropy principle, we introduce *curved neural networks* as a class of models with a limited number of parameters that are particularly well-suited for studying higher-order phenomena. Through exact mean-field descriptions, we show that these curved neural networks implement a self-regulating annealing process that can accelerate memory retrieval, leading to explosive order-disorder phase transitions with multi-stability and hysteresis effects. Moreover, by analytically exploring their memory-retrieval capacity using the replica trick, we demonstrate that these networks can enhance memory capacity and robustness of retrieval over classical associative-memory networks. Overall, the proposed framework provides parsimonious models amenable to analytical study, revealing higher-order phenomena in complex networks.

Complex physical, biological, and social systems often exhibit higher-order interdependencies that cannot be reduced to pairwise interactions between their components[1,2]. Recent studies suggest that higher-order organisation is not the exception but the norm, providing various mechanisms for its emergence[3–6]. Modelling studies have revealed that higher-order interactions (HOIs) underlie collective activities such as bistability, hysteresis, and 'explosive' phase transitions associated with abrupt discontinuities in order parameters[4,7–11].

HOIs are particularly important for the functioning of biological and artificial neural systems. For instance, they shape the collective activity of biological neurons[12,13], being directly responsible for their inherent sparsity[5,13–15] and possibly underlying critical dynamics[16,17]. HOIs have also been shown to enhance the computational capacity of artificial recurrent neural networks[18,19]. More specifically, 'dense associative memories' with extended memory capacity[20–23] are realised by specific non-linear activation functions, which effectively incorporate HOIs. These non-linear functions are related to attention mechanisms

of transformer neural networks[24] and the energy landscape of diffusion models[25,26], leading to the conjecture that HOIs underlie the success of these state-of-the-art deep learning models.

Despite their importance, existent studies of HOIs face significant computational challenges. Analytically tractable models that incorporate HOIs typically limit interactions to a single order (e.g., *p*-spin models[22,27,28]). Otherwise, attempting to represent diverse HOIs exhaustively results in a combinatorial explosion[29]. This issue is pervasive, restricting investigations of high-order interaction models—such as contagion[9], Ising[19], or Kuramoto[30] models—to highly homogeneous scenarios[3,16] or to models of relatively low-order[9,11,31]. While attempts have been made to model all orders of HOIs and perform theoretical analyses[20–23,32–37], it is currently unclear how to construct parsimonious models to address the diverse effects of HOIs in a principled manner.

To address this challenge, here we employ an extension of the maximum entropy principle to capture HOIs through the deformation

[1]BCAM – Basque Center for Applied Mathematics, Bilbao, Spain. [2]IKERBASQUE, Basque Foundation for Science, Bilbao, Spain. [3]Research Division, Araya Inc., Tokyo, Japan. [4]Centre for Complexity Science, Imperial College London, London, UK. [5]Sussex AI and Sussex Centre for Consciousness Science, Department of Informatics, University of Sussex, Brighton, UK. [6]Department of Brain Sciences and Centre for Complexity Science, Imperial College London, London, UK. [7]Center for Eudaimonia and Human Flourishing, University of Oxford, Oxford, UK. [8]Principles of Intelligent Behavior in Biological and Social Systems (PIBBSS), Prague, Czech Republic. [9]Graduate School of Informatics, Kyoto University, Kyoto, Japan. [10]Center for Human Nature, Artificial Intelligence, and Neuroscience (CHAIN), Hokkaido University, Sapporo, Japan. ✉e-mail: maguilera@bcamath.org

of the space of statistical models. When applied to neural networks, our approach generalises classical neural network models to yield a family of *curved neural networks* that effectively incorporate HOIs of all orders. The resulting models have rich connections with the literature on the statistical physics of neural networks[21,22,27,34]. These features enable the exploration of various aspects of HOIs using techniques including mean-field approximations, quenched disorder analyses, and path integrals.

Our analyses reveal how relatively simple curved neural networks exhibit some of the hallmark characteristics of higher-order phenomena, such as explosive phase transitions, arising both in mean-field models and in more complex transitions to spin-glass states. These phenomena are driven by a self-regulated annealing process, which accelerates memory retrieval through positive feedback between energy and an 'effective' temperature—a perspective that can also explain memory-retrieval dynamics in other modern artificial networks. Furthermore, we show—both analytically and experimentally—that this mechanism can lead to an increase in the memory capacity or robustness of memory retrieval in these neural networks. Overall, the core contributions of this work are (i) the development of a parsimonious neural network model based on the maximum entropy principle that captures interactions of all orders, (ii) the discovery of a self-regulated annealing mechanism that can drive explosive phase transitions, and (iii) the demonstration of enhanced memory capacity resulting from this mechanism.

## Results

### High-order interactions in curved manifolds

The maximum entropy principle (MEP) is a general modelling framework based on the principle of adopting the model with maximal entropy compatible with a given set of observations, under the rationale that one should not assume any structure beyond what is specified by the assumptions or features selected from the data[38,39]. The traditional formulation of the MEP is based on Shannon's entropy[40], and the resulting models correspond to Boltzmann distributions of the form $p(\boldsymbol{x}) = \exp\left(\sum_a \theta_a f_a(\boldsymbol{x}) - \varphi\right)$, where $\boldsymbol{x} = (x_1, ..., x_n)$, $\varphi$ is a normalising potential, and $\theta_a$ are parameters constraining the average value of observables $\langle f_a(\boldsymbol{x}) \rangle$. While observables are often set to low orders (e.g. $f_i(\boldsymbol{x}) = x_i$, $f_{ij}(\boldsymbol{x}) = x_i x_j$, corresponding to first and second order statistics), higher-order interdependencies can be included by considering observables of the type $f_I(\boldsymbol{x}) = \prod_{i \in I} x_i$, where $I$ is a set of indices of order $k = |I|$. Unfortunately, an exhaustive description of interactions up to order $k \gg 1$ becomes unfeasible in practice due to an exponential number of terms (for more details on the MEP, see Supplementary Note 1).

The MEP can be expanded to include other entropy functionals such as Tsallis'[41] and Rényi's[42]. Concretely, maximising the Rényi entropy (with the scaling parameter $\gamma \geq -1$)[43]

$$H_\gamma(p) = -\frac{1}{\gamma} \ln \sum_{\boldsymbol{x}} p(\boldsymbol{x})^{1+\gamma} \qquad (1)$$

while constraining $\langle f_a(\boldsymbol{x}) \rangle$ (i.e., the expectation of features by $p(\boldsymbol{x})$) results in models of the form (see Supplementary Note 1):

$$p_\gamma(\boldsymbol{x}) = \exp(-\varphi_\gamma)\left[1 + \gamma\beta \sum_a \theta_a f_a(\boldsymbol{x})\right]_+^{1/\gamma}, \qquad (2)$$

where $\varphi_\gamma$ is a normalising constant given by

$$\varphi_\gamma = \ln \sum_{\boldsymbol{x}}\left[1 + \gamma\beta \sum_a \theta_a f_a(\boldsymbol{x})\right]_+^{1/\gamma}. \qquad (3)$$

Above, the square bracket operator sets negative values to zero, $[x]_+ = \max\{0, x\}$. We refer to distributions following (2) as the *deformed exponential family*, which maximises both Rényi and Tsallis entropies[44,45]. When $\gamma \to 0$, Rényi's entropy tends to Shannon's and (2) to the standard exponential family[42].

A fundamental insight explored in this study is that higher-order interdependencies can be efficiently captured by deformed exponential family distributions[46,47]. Starting from a standard Shannon's MEP model with low-order interactions, it can be shown that varying $\gamma$ in (2) results in a deformation of the statistical manifold which, in turn, enhances the capability of $p_\gamma(\boldsymbol{x})$ to account for higher-order interdependencies. In effect, the consequence of deformation can be investigated by rewriting (2) via Taylor expansion of the exponent

$$p_\gamma(\boldsymbol{x}) = \exp\left(\sum_{k=1}^{\infty} \frac{-1}{k\gamma}\left(-\gamma\beta \sum_a \theta_a f_a(\boldsymbol{x})\right)^k - \varphi_\gamma\right), \qquad (4)$$

which is valid for the case $1 + \gamma\sum_a\theta_a f_a(\boldsymbol{x}) > 0$, and otherwise $p_\gamma(\boldsymbol{x}) = 0$. This shows that the deformed manifold contains interactions of all orders even if $f_a(\boldsymbol{x})$ is restricted to lower orders while establishing a specific dependency structure across the orders, thereby avoiding a combinatorial explosion of the number of required parameters. The deformation resulting from the maximisation of a non-Shannon entropy has been shown to reflect a curvature of the space of possible models in information geometry[42,45,48,49]. This leads to a particular *foliation* of the space of possible models[50] (an 'onion-like' manifold structure, Fig. 1), which has properties that allow to re-derive the MEP from fundamental geometric properties—for technical details, see Supplementary Note 1.

### Curved neural networks

Several well-known neural network models adhere to the MEP, such as Ising-like models[51] and Boltzmann machines[52]. Interestingly, these models can encode patterns in their weights in the form of 'associative memories' as in Nakano-Amari-Hopfield networks[53–55], being amenable for investigations using tools from equilibrium and nonequilibrium statistical physics literature[56–59]. Following the principles laid down in the previous section, we now introduce a family of recurrent neural networks that we call *curved neural networks*.

For this purpose, let us consider $N$ binary variables $x_1, ..., x_N$ taking values in $\{1, -1\}$ following a joint probability distribution

$$p_\gamma(\boldsymbol{x}) = \exp(-\varphi_\gamma)[1 - \gamma\beta E(\boldsymbol{x})]_+^{1/\gamma}, \qquad (5)$$

where $\varphi_\gamma$ is a normalising constant. Above, we call $E(\boldsymbol{x})$ and $\beta$ the (stochastic) *energy function* (i.e., Hamiltonian) and the *inverse temperature*, due to their similarity with the Gibbs distribution in statistical physics when $\gamma \to 0$. Note that, unlike exponential families, these models do not exhibit energy invariance under constant shifts. However, as demonstrated in Ref. 41, deformed exponential models can be related to energy-invariant models by rescaling their temperature, which can be seen as maximising entropy with respect to escort statistics rather than the original natural statistics.

Neural network models are typically defined by considering $p_\gamma(\boldsymbol{x})$ as defined in (5) with an energy function of the form

$$E(\boldsymbol{x}) = -\sum_{i=1}^{N} H_i x_i - \frac{1}{N}\sum_{i<j} J_{ij} x_i x_j, \qquad (6)$$

where $J_{ij}$ is the coupling strength between neurons $x_i$ and $x_j$, and $H_i$ are bias terms. In the limit $\gamma \to 0$, $p_0(\boldsymbol{x})$ recovers the Ising model. Emulating classical associative memories, the weights $J_{ij}$ can be made to encode a collection of $M$ neural patterns $\boldsymbol{\xi}^a = \{\xi_1^a, ... \xi_N^a\}$, $\xi_1^a = \pm 1$ and $a = 1, ..., M$

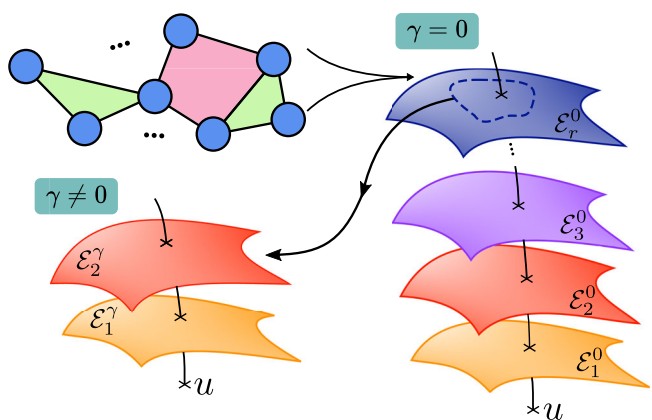

**Fig. 1 | Higher-order decomposition resulting from the foliation of a statistical manifold.** Illustration of a family of standard MEP models (right) and its deformed counterpart (bottom left). The space of MEP distributions with constraints of different orders constitute nested sub-manifolds[29], giving rise to a hierarchy of sub-families of models of the form $\mathcal{E}_k^\gamma = \{p_\gamma^{(k)}(\boldsymbol{x}) = e^{-\varphi_\gamma}[1 - \gamma \beta E_k(\boldsymbol{x})]_+^{1/\gamma}\}$ such that $\mathcal{E}_1^\gamma \subset \mathcal{E}_2^\gamma \subset \cdots \subset \mathcal{E}_n^\gamma$ [42]. The foliation depends on the curvature $\gamma$, and in general $\mathcal{E}_k^\gamma \neq \mathcal{E}_k^0$ but rather $\mathcal{E}_k^\gamma \cap \mathcal{E}_r^0 \neq \varnothing$ for $k < r$. For small values of $|\gamma|$, it is possible to neglect higher-order terms in (4), and therefore certain subsets of $\mathcal{E}_k^\gamma$ effectively approximate $\mathcal{E}_r^0$.

by using the well-known Hebbian rule[55,56]

$$J_{ij} = J \sum_{a=1}^{M} \xi_i^a \xi_j^a, \tag{7}$$

where $J$ is a scaling parameter.

Before proceeding with our main analysis, one can gain insights into the effect of the curvature $\gamma$ from the dynamics of a recurrent neural network that behaves as a sampler of the equilibrium distribution described by (5). For this, we adapt the classic Glauber dynamics to curved neural networks (see Supplementary Note 2) to obtain

$$p(x_i | \boldsymbol{x}_{\backslash i}) = \left(1 + [1 - \gamma \beta'(\boldsymbol{x}) \Delta E(\boldsymbol{x})]_+^{1/\gamma}\right)^{-1}, \tag{8}$$

where $\boldsymbol{x}_{\backslash i}$ denotes the state of all neurons except $x_i$, $\Delta E(\boldsymbol{x}) = 2x_i(H_i + \frac{1}{N}\sum_j J_{ij} x_j)$ is the energy difference associated with detailed balance, and $\beta'(\boldsymbol{x})$ is an effective inverse temperature given by

$$\beta'(\boldsymbol{x}) = \frac{\beta}{[1 - \gamma \beta E(\boldsymbol{x})]_+}. \tag{9}$$

Again, $\gamma \to 0$ recovers the classic Glauber dynamics and $\beta'(\boldsymbol{x}) = \beta$. Thus, the curvature affects the dynamics through the deformed nonlinear activation function (8) and the state-dependent effective temperature $\beta'(\boldsymbol{x})$ (9), with higher $\beta'(\boldsymbol{x})$ inducing lower degrees of randomness in the transitions. The effect of $E(\boldsymbol{x})$ on $\beta'(\boldsymbol{x})$ depends then on the sign of $\gamma$. A negative $\gamma$ increases $\beta'(\boldsymbol{x})$ during relaxation, reducing the stochasticity of the dynamics and accelerating convergence to a low-energy state. This, in turn, raises $\beta'$, creating a positive feedback loop between energy and effective temperature. The effect is similar to simulated annealing, but the coupling of the energy and effective inverse temperature lets the annealing scheduling self-regulate to accelerate convergence. In contrast, positive $\gamma$ decelerates the dynamics through negative feedback. Such accelerating or decelerating dynamics underlie non-trivial complex collective behaviours of the curved neural networks, which will be examined in the subsequent sections.

## Mean-field behaviour of curved associative-memory networks

As with regular associative memories[58], one can solve the behaviour of curved associative-memory networks through mean-field methods in the thermodynamic limit $N \to \infty$ (Supplementary Note 3). Here the energy is extensive, meaning that it scales with the system's size $N$. To ensure the deformation parameter remains independent of system properties such as size or temperature, we scale it as follows:

$$\gamma = \frac{\gamma'}{N\beta}. \tag{10}$$

Under this condition, we calculate the normalising potential $\varphi_\gamma$ by introducing a delta integral and calculating a saddle-node solution, resulting in a set of order parameters $\boldsymbol{m} = \{m_1, \ldots, m_M\}$, $m_a = \frac{1}{N}\sum_i \xi_i^a \langle x_i \rangle$ in the limit of size $N \to \infty$. This calculation assumes $1 - \gamma \beta E(\boldsymbol{x}) > 0$ so that $[]_+$ operators can be omitted and $\varphi_\gamma$ is differentiable. The solution results in (for $H_i = 0$):

$$\varphi_\gamma = N \frac{\beta}{\gamma'} \ln \frac{\beta'}{\beta} - \sum_{a=1}^{M} \beta' N J m_a^2 + \sum_{i=1}^{N} \ln \left(2 \cosh \left(\beta' J \sum_{a=1}^{M} \xi_i^a m_a\right)\right), \tag{11}$$

where $\beta'$ is given by

$$\beta' = \frac{\beta}{1 + \gamma' \frac{1}{2} J \sum_a m_a^2}, \tag{12}$$

and the values of the mean-field variables $m_a$ are found from the following self-consistent equations:

$$m_a = \sum_{i=1}^{N} \frac{\xi_i^a}{N} \tanh \left(\beta' J \sum_{b=1}^{M} \xi_i^b m_b\right). \tag{13}$$

Similarly, using a generating functional approach[59], we use the Glauber rule in (8) to derive a dynamical mean-field given by path integral methods (see Supplementary Note 4). This yields

$$\dot{m}_a = -m_a + \sum_{i=1}^{N} \frac{\xi_i^a}{N} \tanh \left(\beta' J \sum_{b=1}^{M} \xi_i^b m_b\right), \tag{14}$$

where $\beta'$ is defined as in (12) for each $\boldsymbol{m}$. Note that in large systems, we recover the classical nonlinear activation function, and the deformation affects the dynamics only through the effective temperature $\beta'$.

## Explosive phase transitions

To illustrate these findings, let us focus on a neural network with a single associative pattern ($M = 1$), which is similar to the Mattis model[60] and equivalent to a homogeneous mean-field Ising model[61] (with energy $E(\boldsymbol{x}) = -\frac{1}{N}J\sum_{i<j} x_i x_j$) by changing a variable $x_i \leftarrow \xi_i x_i$. Rewriting (13), we find that a one-pattern curved neural network follows a mean-field model given by

$$m = \tanh(\beta' J m), \tag{15}$$

$$\beta' = \frac{\beta}{1 + \gamma' \frac{1}{2} J m^2}. \tag{16}$$

This result generalises the well-known Ising mean-field solution $m = \tanh(\beta J m)$, which is recovered for $\gamma = 0$.

By evaluating these equations, one finds that the model exhibits the usual order-disorder phase transition for positive and small

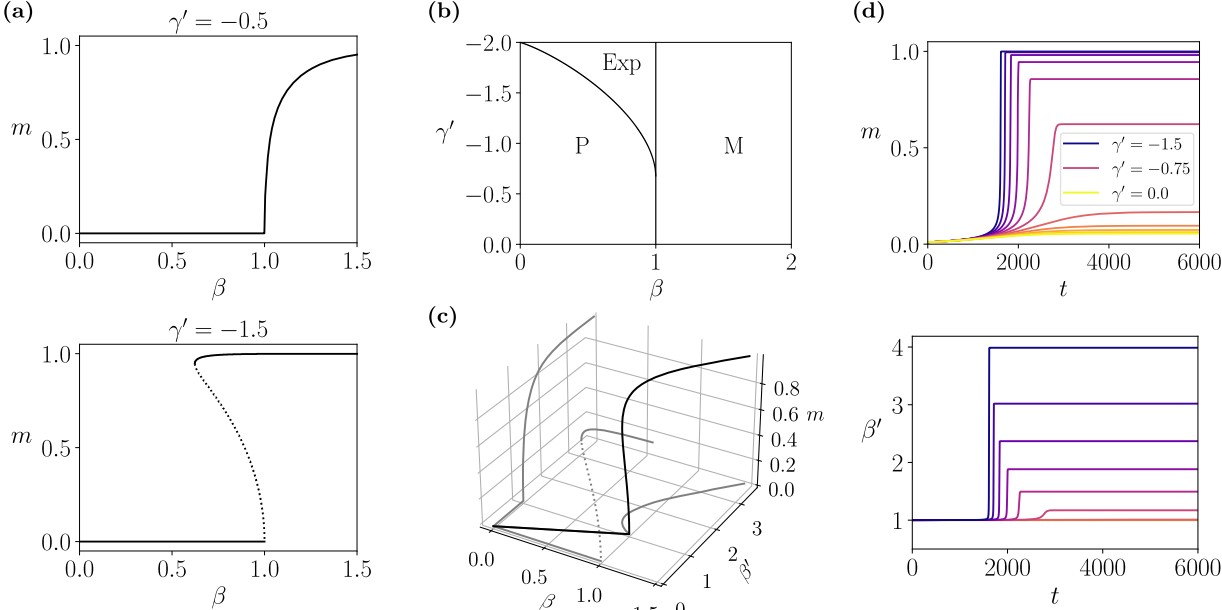

**Fig. 2 | Explosive phase transitions in curved neural networks. a** Phase transitions of the curved neural network with one associative memory, for $J = 1$ and values of $\gamma' = -0.5$ (top, displaying a second-order phase transition) and $\gamma' = -1.5$ (bottom, displaying an explosive phase transition). Solid lines represent the stable fixed points, and dotted lines correspond to unstable fixed points. **b** Phase diagram of the system. The areas indicated by P and M refer to the usual paramagnetic (disordered) and magnetic (ordered) phases, respectively. The area indicated by Exp represents a phase where ordered and disordered states coexist in an explosive phase transition characterised by a hysteresis loop. (**c**) Solutions of (15)-(16) for $\beta', m, \beta$ (black line) for $\gamma' = -1.2$, and projections to the plane $m = 0, \beta = 0$ and $\beta' = 0$, obtaining respectively the relation between $\beta, \beta'$ and solutions of the flat and the deformed models respectively (grey lines). (**d**) Mean-field dynamics of the single-pattern neural network for $\beta = 1.001$ (near criticality from the ordered phase) for some values of $\gamma'$ in [ −1.5, 0]. For large negative $\gamma'$ the dynamics 'explodes', with $m$ (top) and $\beta'$ (bottom) converging abruptly.

negative values of $\gamma'$ (Fig. 2a top). However, for large negative values of $\gamma'$, a different behaviour emerges: an explosive phase transition[8] that displays hysteresis due to HOIs (Fig. 2a bottom). The resulting phase diagram (Fig. 2b) closely resembles phase transitions in higher-order contagion models[9,11] and higher-order synchronisation observed in Kuramoto models[30].

One can intuitively interpret the effect of the deformation parameter $\gamma'$ by noticing that, for a fixed $\beta'$, $m$ is the solution of a function of $\beta'$. For $\gamma' = 0$, this results in the mean-field behaviour of the regular exponential model, which assigns a value of $m$ to each inverse temperature $\beta = \beta'$. In the case of the deformed model, the possible pairs of solutions $(m, \beta')$ are the same, but their mapping to the inverse temperatures $\beta$ changes. Namely, this deformation can be interpreted as a stretching (or contraction) of the effective temperature, which maps each pair $(m, \beta')$ to an inverse temperature $\beta = \beta'(1 + \frac{1}{2}\gamma' J m^2)$ according to (16). Thus, one can obtain the mean-field solutions of the deformed patterns as mappings of the solutions of the original model. This is illustrated in Fig. 2c, where the solution of $\beta', m, \beta$ is projected to the planes $\beta = 0$ and $\beta' = 0$, obtaining the solutions for the flat ($\gamma' = 0$) and the deformed ($\gamma' = -1.2$) models respectively.

In order to gain a deeper understanding of the explosive nature of this phase transition, we study the dynamics of the single-pattern neural network. By rewriting (14) for $M = 1$, and under the change of variables mentioned above to remove $\xi$, the dynamical mean-field equation of the system reduces to

$$\dot{m} = -m + \tanh(\beta' J m), \tag{17}$$

where $\beta'$ is calculated as in (16). Simulations of the dynamical mean-field equations for values of $\beta$ just above the critical point are depicted in Fig. 2d. Trajectories with strongly negative $\gamma'$ saturate earlier than smaller negative $\gamma'$, confirming accelerated convergence. During this process, the effective inverse temperature $\beta'$ rapidly increases until it

saturates, creating a positive feedback loop between $\beta'$ and $m$ that gives rise to the explosive nature of the phase transition. This positive loop occurs only if $\gamma'$ is negative; otherwise, negative feedback simply makes the convergence of $m$ slower.

## Overlaps between memory basins of attraction
A key property of associative-memory networks is their ability to retrieve patterns in different contexts. In the case of one-pattern associative-memory networks, the energy function $E(\boldsymbol{x}) = -\frac{J}{N}\sum_{i<j} x_i \xi_i \xi_j x_j$ is a quadratic function with two minima at $\boldsymbol{x} = \pm \boldsymbol{\xi}$, which configure global attractors. Instead, a two-pattern associative-memory network has an energy function with four minima (if sufficiently separated), but their attraction basins can overlap when the patterns are correlated.

To study the degree of the overlap between pairs of patterns, we analyse solutions of (13) for a network with two patterns with correlation $\langle \xi_i^1 \xi_i^2 \rangle = C$ (see Supplementary Note 3.3 for details). In this scenario, the system is described by two mean-field patterns:

$$\begin{aligned} m_a = &\frac{1}{2}(1+C)\tanh\big(\beta' J(m_1 + m_2)\big) \\ &+ w\frac{1}{2}(1-C)\tanh\big(\beta' J(m_1 - m_2)\big) \end{aligned} \tag{18}$$

with $w = 3 - 2a = \pm 1$ for $a = 1, 2$, and

$$\beta' = \frac{\beta}{1 + \gamma\frac{1}{2}J(m_1^2 + m_2^2)}. \tag{19}$$

Figure 3 shows how the hysteresis effect and explosive phase transitions persist in the case of two patterns for $C = 0.2$ with negative $\gamma'$. This example shows two consecutive, overlapping explosive bifurcations (going from 1 to 2, and then to 4 fixed points), creating a hysteresis involving 7 fixed points within a more compressed

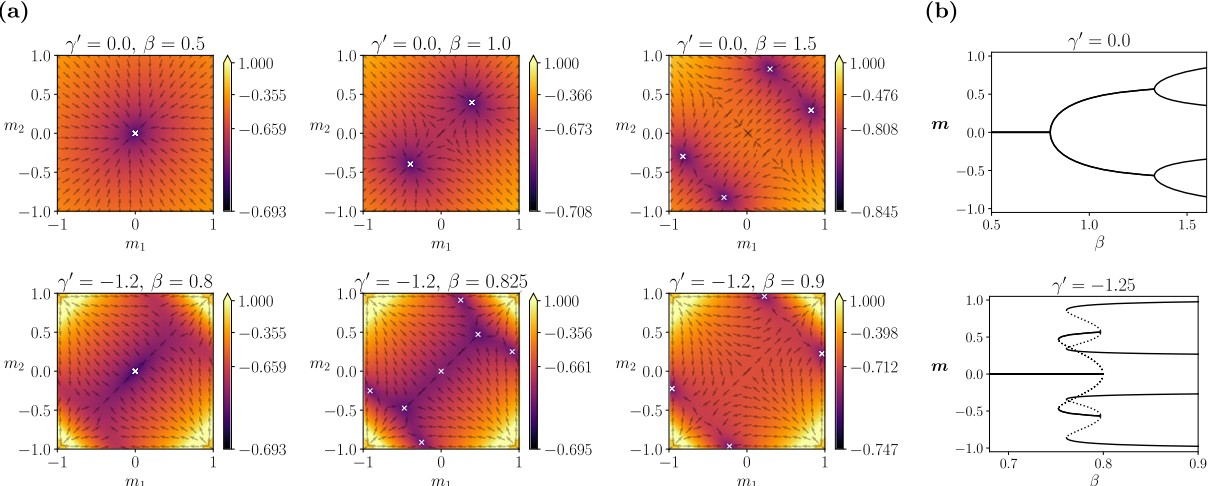

**Fig. 3 | Interaction between two encoded memories. a** Values of $\varphi_\gamma$ for different mean-field values $m_1$, $m_2$, indicating the attractor structure of the network for different values of $\beta$ with $J = 1$, $C = 0.2$ for $\gamma' = 0$ (top row) and $\gamma' = -1.2$ (bottom row). **b** Bifurcations of the order parameters $m_1$, $m_2$. For $\gamma' = 0$ we observe an attractor bifurcating into two and then into four. For $\gamma' = -1.2$, we observe the same sequence, but with a coexistence hysteresis regime in which 7 attractors are possible.

parameter range of $\beta$ than the classical case. Consequently, the memory-retrieval region for the four embedded memories expands. These results illustrate complex hysteresis cycles as well as an increased memory capacity for finite temperatures by negative values of $\gamma'$. This enhanced capability for memory retrieval is further investigated through the replica analyses in the next section.

## Memory retrieval with an extensive number of patterns

Next, we investigate how the deformation related to $\gamma$ impacts the memory-storage capacity of associative memories. In classical associative networks of $N$ neurons, the energy function is defined as $E(\boldsymbol{x}) = -\frac{J}{N}\sum_{a=1}^{M}\sum_{i<j} x_i \xi_i^a \xi_j^a x_j$ with $M = \alpha N$. As the number of patterns learned by the network increases, the system transitions to a disordered spin-glass state in the thermodynamic limit. Furthermore, one can analytically solve this model[62–65]. For example, using the replica-trick method can determine the memory capacity of the system[62], and theoretically identify the critical value of $\alpha$ at which memory retrieval becomes impossible–leading to a disordered spin-glass phase. Here, we apply a similar approach to reveal how deformed associative memory networks afford an enhanced memory capacity.

Applying the replica trick in conjunction with the methods outlined in previous sections allows us to solve the system (see Supplementary Note 5). This method entails computing a mean-field variable $m$ corresponding to one of the patterns $\boldsymbol{\xi}^a$ and averaging over the others. For simplicity, a pattern with all positive unity values $\boldsymbol{\xi}^a = (1, 1, ..., 1)$ is considered, which is equivalent to any other single pattern just by a series of sign flip variable changes. The degree of similarity or overlap of this pattern with other patterns in the system introduces a new order parameter $q$, which contributes to measuring disorder in the system. After introducing the relevant order parameters and solving under a replica-symmetry assumption, the normalising potential is derived as

$$
\begin{aligned}
\varphi_\gamma = & N\frac{\beta}{\gamma'}\ln\frac{\beta}{\beta'} - N\beta'Jm^2 - N\frac{1}{2}\alpha(\beta'J)^2(r + R - 2qr) \\
& - N\frac{1}{2}\alpha\left(\ln(1 - \beta'J(1-q)) - \beta'J\sqrt{rq}\right) \\
& + N\int Dz \ln\left(2\cosh(\beta'Jm + \beta'J\sqrt{\alpha r}z)\right),
\end{aligned}
\tag{20}
$$

where $J$ is a scaling factor, and the order parameters are defined as

$$
m = \int Dz\, \tanh(\beta'Jm + \beta'J\sqrt{\alpha r}z),
\tag{21}
$$

$$
q = \int Dz\, \tanh^2(\beta'Jm + \beta'J\sqrt{\alpha r}z),
\tag{22}
$$

with

$$
r = \frac{q}{(1 - \beta'J(1-q))^2}, \quad R = \frac{(\beta'J)^{-1} - (1 - 2q)}{(1 - \beta'J(1-q))^2}.
\tag{23}
$$

As in previous cases, the model is governed by an effective temperature

$$
\beta' = \frac{\beta}{1 + \gamma'\frac{1}{2}\left(Jm^2 + \alpha J(\beta'(R - qr) - 1)\right)}.
\tag{24}
$$

This solution differs from the models in previous sections by the self-dependence of $\beta'$.

To obtain a phase diagram, we solved (21)-(22) numerically for given $\alpha$, $\beta'$ at $\gamma' = 0$, and rescaled the inverse temperature as in the previous section to obtain the corresponding values of $\beta$ for each $\gamma'$. Using the resulting order parameters and calculating the free energy for each $\alpha$, $\beta$, $\gamma'$, we constructed the phase diagram of the system (similarly to regular associative memories[58,62]) characterised by the following distinct phases (Fig. 4):

- A paramagnetic phase (P), corresponding to disordered solutions with $m = q = 0$, where memory-retrieval fails due to the dominance of fluctuations.
- A ferromagnetic phase (F), corresponding to stable memory-retrieval solutions with $m > 0$ and $q > 0$.
- A spin-glass phase (SG), exhibiting spurious-retrieval solutions with $m = 0$ and $q > 0$.
- A mixed phase (M), where F and SG types of solutions coexist, being the spin-glass solutions a global minimum of the normalising potential $\varphi_\gamma$.

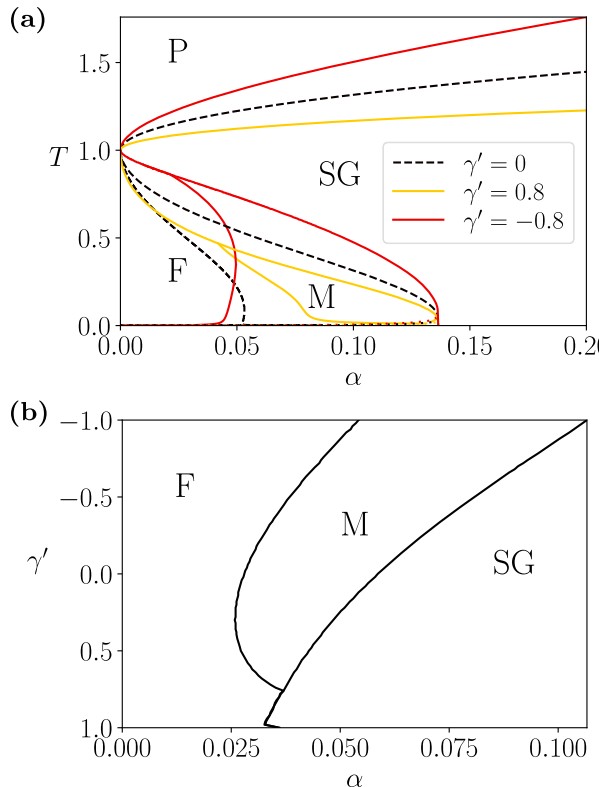

**(a)**

**(b)**

**Fig. 4 | Memory capacity is enhanced by geometric deformation.** Phase diagram of a curved associative memory with an extensive number of encoded patterns $M = \alpha N$ and $J = 1$ for (**a**) different $T = 1/\beta$ at $\gamma' = 0$ (black dashed lines), 0.8, $-$0.8 (solid lines), and for (**b**) different $\gamma'$ at $\beta = 2$. F indicates the ferromagnetic (i.e., memory retrieval) phase, SG the spin-glass phase (where saturation makes memory retrieval inviable), M a mixed phase, and P the paramagnetic region. Both in F and M, ferromagnetic and spin-glass solutions coexist, but we differentiate these by calculating respectively whether memory-retrieval or spin-glass solutions are the global minimum of the normalising potential $\varphi_\gamma$. The dotted lines in (a) near $T = 0$ indicate the AT lines, below which the replica-symmetric solution is not valid. Increasing $\gamma'$ to larger negative values extends the retrieval phase into larger values of $\alpha$, indicating an increased memory capacity, while larger positive values reduce the extension of the mixed phase, increasing robustness of memory retrieval.

For $\gamma' = 0$ (black dashed lines), the phase transition reflects the behaviour of associative memories near saturation[58,62]. With negative $\gamma'$ (red lines), we observe an expansion of the ferromagnetic and mixed phases, indicating an enhanced memory-storage capacity by the deformation. Conversely, a positive value of $\gamma'$ (yellow lines) decreases the memory capacity but reduces the extent of the mixed phase. In the mixed phase, retrieved memories ($m > 0$) are represented at a local—but not global—minimum of the normalising potential $\varphi_\gamma$ in (20), indicating a larger probability of observing spurious patterns. Thus, we expect positive values of $\gamma'$ to result in more robust memory retrieval.

The stability of the replica symmetry solution is given by the condition

$$\left(1 + \beta'(1 - q)\right)^2 > \alpha \beta'^2 \int Dz \cosh^{-4} \beta' \left(Jm + J\sqrt{\alpha r}z\right), \qquad (25)$$

which is captured by the dotted lines near zero temperature in Fig. 4a. Note that all solutions in Fig. 4b are stable under the replica symmetry assumption.

We complement the analysis from the previous section with an experimental study of a system encoding patterns from an image classification benchmark. The patterns are sourced from the CIFAR-

100 dataset, which comprises 60,000 32 × 32 colour images[66]. To adapt the dataset to binary patterns suitable for storage in an associative memory, we processed each RGB channel by assigning a value of 1 to pixels with values greater than the channel's median value and −1 otherwise (Fig. 5a). The resulting array of $N = 32 \cdot 32 \cdot 3$ binary values for each image was assigned to patterns $\xi^a$. Note that associative memories (as well as our theory above) usually assume that patterns are relatively uncorrelated, and specific methods are required to adapt them to correlated patterns[67,68]. To simplify the problem, we conducted experiments using a selection of 100 images with covariance values smaller than $10/\sqrt{N}$ (the standard deviation of the covariance values for uncorrelated patterns is $1/\sqrt{N}$). We used a random search to select patterns with low correlations: we randomly picked an image and replaced it if its correlation exceeded the threshold, repeating until all correlations were below it.

We evaluated the memory retrieval capacity of networks with various degrees of curvature $\gamma$ by encoding different numbers of memories, as described in (7). As a measure of performance, we evaluated the stability of the network by assigning an initial state $x = \xi^a$ and calculating the overlap $o = \sum_i x_i \xi_i^a$ after $T = 30N$ Glauber updates for $\beta = 2, J = 1$. The process was repeated $R = 500$ times from different initial conditions (different encoded patterns and different initial states) to estimate the value of $m$ in (21). Experimental outcomes confirm our theoretical results, revealing that memory capacity increases with negative values of $\gamma'$, while positive values reduce the memory capacity (Fig. 5b), but reduce the extent and magnitude of the high variability region in pattern retrieval (Fig. 5c), which is consistent with the reduction of the mixed phase. Note that the resulting memory capacity of the system observed in our experiments (i.e., the value of $\alpha$ at which the transition happens) is diminished due to the presence of correlations among some of the memorised patterns.

Finally, we investigated transitions near the spin-glass phase boundaries. First, we note that, for $J \to 0$ and $\alpha = J^{-2}$, the model in (21)-(22) converges to (see Supplementary Note 5)

$$q = \int Dz \tanh^2 \left(\beta' \sqrt{q}z\right), \qquad (26)$$

$$\beta' = \frac{\beta}{1 + \frac{1}{2}\gamma'\beta'(1 - q^2)}, \qquad (27)$$

which at $\gamma = 0$ recovers the well-known Sherrington-Kirkpatric model[69] (see Supplementary Note 6). While in the classical case, a phase transition occurs from a paramagnetic to a spin-glass phase, the curvature effect of $\gamma' \neq 0$ modifies the nature of this transition. For small values of $\gamma'$, the system exhibits a continuous phase transition akin to the Sherrington–Kirkpatrick spin-glass, where $\frac{dq}{d\beta}$ shows a cusp (Fig. 6a). However, for $\gamma' = -1$ the phase transition becomes second-order, displaying a divergence of $\frac{dq}{d\beta}$ at the critical point (Fig. 6b). Moreover, increasing the magnitude of negative $\gamma'$ leads to a first-order phase transition with hysteresis (Fig. 6c), resembling the explosive phase transition observed in the single-pattern associative-memory network. This hybrid phase transition combines the typical critical divergence of a second-order phase transition with a genuine discontinuity, similar to 'type V' explosive phase transitions[8].

We analytically calculated the properties of these phase transitions (see Supplementary Note 6). By computing the solution at $\gamma' = 0$ and rescaling $\beta'$, we determined that the critical point is located at $\beta_c = 1 + \frac{1}{2}\gamma'$ (consistent with Fig. 6a–c). The slope of the order parameter around the critical point is, for $\gamma' \leq -1$, equal to $(1 + \gamma')^{-1}$, indicating the onset of a second-order phase transition as depicted in Fig. 6b. The resulting phase diagram of the curved Sherrington-Kirkpatrick model is shown in Fig. 6d.

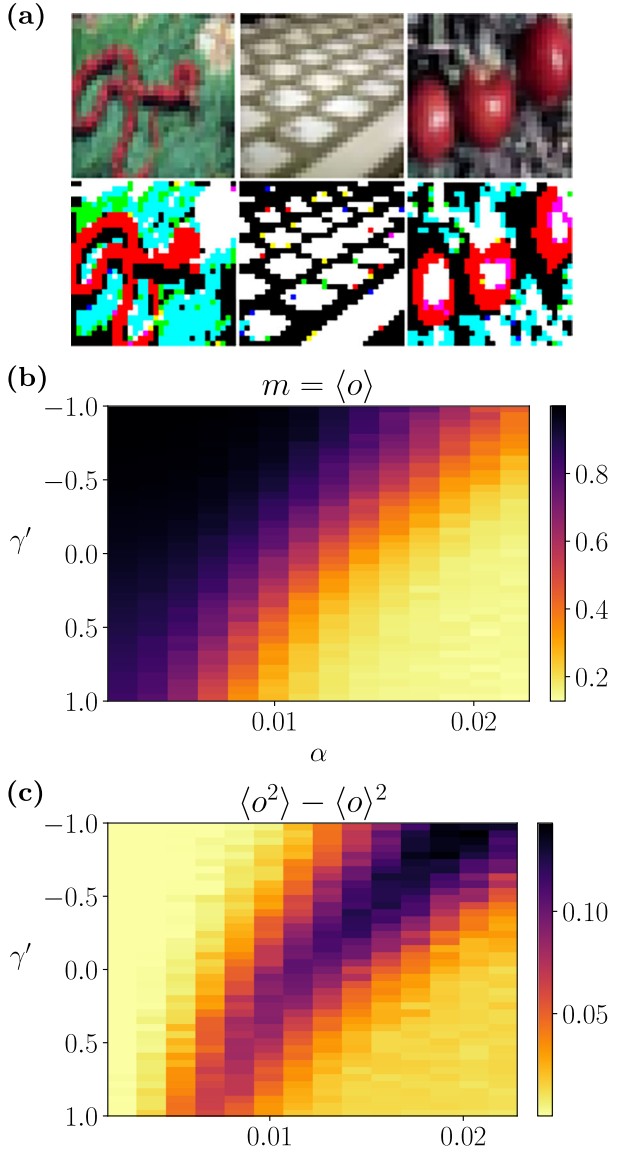

**Fig. 5 | Simulation study for the effect of deformation on image encoding.**
**a** Examples of CIFAR-100 images (top) and their RGB binarised versions (bottom).
Every 32 × 32 × 3 binary RGB pixel value for each image $a$ is assigned to the value of
one position of pattern $\xi_i^a$. **b, c** Mean and variance of pattern retrieval values
obtained in experiments, measured by the overlap between the final state of the
network and the encoded pattern.

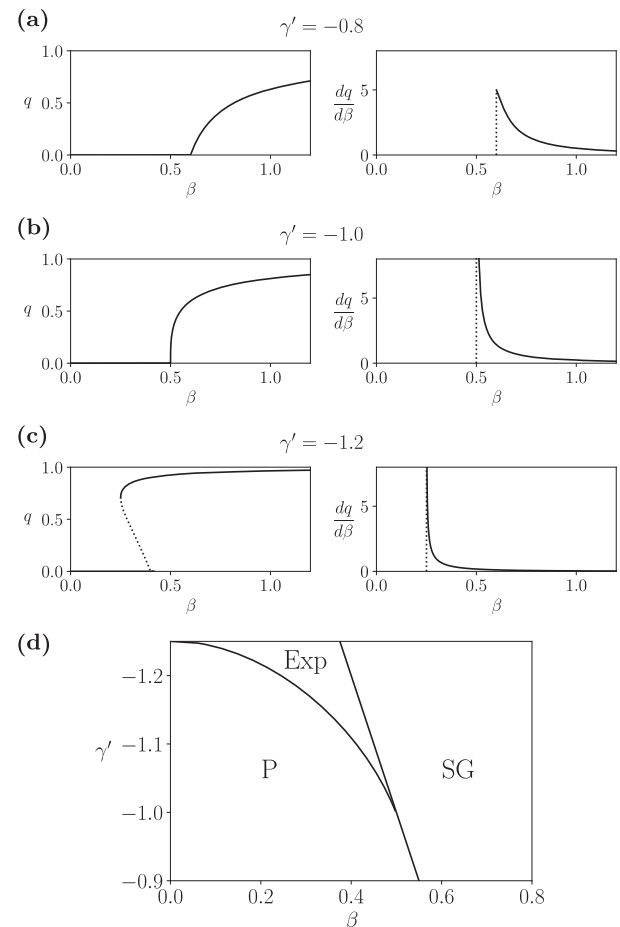

**Fig. 6 | Explosive spin glasses.** Phase transitions for order parameter $q$ for replica-
symmetric disordered spin models displaying (**a**) a cusp phase transition for
$\gamma' = -0.5$, (**b**) a second-order phase transition for $\gamma' = -1.0$ and (**c**) an explosive
phase transition for $\gamma' = -1.2$. **d** Phase diagram of the explosive spin glass, dis-
playing a paramagnetic (P), spin-glass (SG) and an explosive phase (Exp).

## Comparison with other dense associative memory models

Although our primary objective is to develop a parsimonious model of
HOIs to explain higher-order phenomena, our framework can also be
used to explain the behaviour of modern networks with HOIs,
including the recently proposed relativistic Hopfield model[32–34] and
dense associative memories[20,21]. For this, let us consider the energy
$\mathcal{F}[E]$ of the exponential family distribution $p(\boldsymbol{x}) \sim e^{-\beta\mathcal{F}[E]}$ given by the
nonlinear transformation (denoted by $\mathcal{F}$) of the classical energy $E(\boldsymbol{x})$.
The deformed exponential models in this study correspond to
$\mathcal{F}[E] = -\frac{N}{\gamma'}\ln(1 - \gamma' E/N)$, while the relativistic model corresponds to
$\mathcal{F}[E] = -\frac{N}{\gamma'}\sqrt{1 - \gamma' E/N}$. For the deformed exponential, the term $\mathcal{F}[E]$
can be expanded as

$$\mathcal{F}[E] = E + \frac{\gamma'}{2N}E^2 + \frac{\gamma'^2}{3N^2}E^3 + \dots \tag{28}$$

When $E$ depends on the quadratic Mattis magnetisation (i.e.,
$E = -\sum_a \frac{1}{N}\left(\sum_i \xi_i^a x_i\right)^2$), then $\mathcal{F}[E]$ expands in terms of even-order HOIs
of $\sum_i \xi_i^a x_i$. For $\gamma' < 0$, all coefficients of $\sum_i \xi_i^a x_i$ in the expansion are
negative, indicating that embedded memories have deeper energy
minima than in the classical case. The same signs appear for each order
in the relativistic energy with $\gamma' < 0$. We also note that $\beta$ in the free
energy of both the deformed exponential and relativistic models in the
limit of large $N$ appears scaled according to an effective temperature
given by $\beta' = \beta\partial_E \mathcal{F}[E]$ (e.g., (11) and Eq. (6.2) in Ref. 34). Moreover, the
input in the Glauber dynamics is approximated for large sizes as

$$\beta\Delta\mathcal{F}[E] \approx \beta\partial_E\mathcal{F}[E]\,\Delta E(\boldsymbol{x}) = \beta'\Delta E(\boldsymbol{x}). \tag{29}$$

The effective inverse temperatures $\beta' = \beta(1 - \gamma' E/N)^{-1}$ for the
deformed exponential and $\beta' = 2^{-1}(1 - \gamma' E/N)^{-1/2}$ for the relativistic
models are decreasing functions of $E$ when $\gamma' < 0$, resulting in an
acceleration of memory retrieval—with lower energy $E$ resulting in
higher $\beta'$ (lower temperature). While the relativistic model has been
studied for $\gamma' > 0$[32–34], we conjecture it may exhibit explosive phase
transitions if $\gamma' < 0$. Conversely, a positive $\gamma'$ introduces alternating
signs in even-order terms of $\sum_i \xi_i^a x_i$, and a shallower energy
landscape due to a reduction in $\beta'$. This shallower energy landscape reduces the
memory capacity of the deformed exponential networks by expanding
the spin-glass phases (Fig. 4), but also enlarges the recall (ferromag-
netic) region by mitigating the formation of spurious memories given

by overlapping patterns in the mixed phase (in alignment with previous work[32] on mitigation of spurious memories in the relativistic model).

This perspective on accelerated memory retrieval by nonlinearity extends to dense associative memories[20,21], which achieve supralinear memory capacities through nonlinear pattern encoding. Specifically, their energy function is given by $\mathcal{F} = -\sum_a F(\sum_i \xi_i^a x_i)$ with $F$ being e.g., a thresholded power function[20], $F(z) = [z]_+^p$ or an exponential nonlinearity[21] $F(z) = e^z$ at zero temperature. These nonlinearities narrow basins of attraction, reducing memory overlap and preventing transitions to the spin-glass phase. The jumps in the Glauber dynamics of such systems are weighed by an accelerating function. Namely, from our perspective, the dynamics of such systems can be described via positive feedback on weights linked to a specific memory, which increase during memory retrieval. This follows from the fact that, relating the linear difference in Mattis terms $\Delta\epsilon_k^a \equiv 2\xi_k^a x_k$ with the nonlinear difference $\Delta F_k^a \equiv F(\sum_i \xi_i^a x_i) - F(\sum_i \xi_i^a x_i - \Delta\epsilon_k^a)$, the update of the $k$th neuron is determined by the sign of

$$\Delta\mathcal{F}(\boldsymbol{x}) = \sum_a \frac{\Delta F_k^a}{\Delta\epsilon_k^a}\Delta\epsilon_k^a = \sum_a w_k^a \Delta\epsilon_k^a. \tag{30}$$

Here, we show that the effective weight $w_k^a \equiv \frac{\Delta F_k^a}{\Delta\epsilon_k^a}$ becomes an increasing function of $\sum_i \xi_i^a x_i$ when $F$ is the power, exponential, or more generally, a convex function (See Supplementary Note 7). Thus, increasing $\sum_i \xi_i^a x_i$ as pattern $\boldsymbol{\xi}^a$ is retrieved strengthens its basin of attraction and ensures positive feedback. Meanwhile, retrieval of $\boldsymbol{\xi}^a$ reduces $\sum_i \xi_i^b x_i$ for orthogonal patterns $\boldsymbol{\xi}^b$, lowering their weights, suppressing their recall to minimise interference. This competitive mechanism highlights the higher memory capacity of these models compared to curved neural networks with uniform temperature scaling. Unlike the effective inverse temperature in curved networks, which depends only on the system's state or energy, the effective weight in updating the $k$-th neuron additionally depends on the neuron's state $x_k$, thus no longer representing a global modulation of the energy.

## Discussion

HOIs play a critical role in enabling emergent collective phenomena in natural and artificial systems. Modelling HOIs is, however, highly non-trivial, often requiring advanced analytic tools (such as simplicial complexes or hypergraphs) that entail an exponential increase in parameters for large systems. In this paper, we addressed this issue by leveraging the maximum entropy principle to effectively capture HOIs in models via a deformation parameter $\gamma$, which is associated with the Rényi entropy. Given their close connection with statistical physics, this family of models provides a useful setup to investigate the effect of HOIs on spin systems, including explosive ferromagnetic and spin-glass phase transitions, extending studies on anomalous phase transitions found in other systems[2,7–9,11], and the capability of networks to store memories.

The observed effects in curved neural networks can be explained via an effective temperature, inducing a positive or negative feedback effect in memory retrieval. As we discussed above, this effect is present in different forms across other dense associative memories[20,21,34]. A similar argument may apply to diffusion models framed within dense associative memories[25,26], where the energy follows a log-sum-exp nonlinearity. Thus, the accelerated mechanism found in this study clarifies memory retrieval in advanced associative networks, providing an important step toward designing extended memory capacities and improved noise scheduling.

Curved neural networks also provide insights into biological neural systems, where evidence suggests the presence of alternating positive and negative HOIs for even and odd orders, respectively. This alternation leads to sparse neuronal activity, which has been shown to be instrumental for enabling extended periods of total silence[5,13–15,35]. Interestingly, such sparse activity patterns may coexist with the accelerated memory retrieval dynamics, as both involve positive even-order HOIs. The attainment of enhanced memory, combined with sparse activity, presents a promising direction for understanding energy-efficient biological neuronal networks[35,36]. Future work may investigate how curved neural networks might support both energy efficiency and high memory capacities, potentially by adopting a thresholded, supralinear neuronal activation function[20,35]. Additionally, developing statistical methods for fitting these models to experimental data (i.e., theories for learning) represents an important, yet largely unexplored, research avenue. Together, these research directions offer a compelling path to uncover the principles of efficient information coding in biological neural systems.

Overall, our results demonstrate the benefits of considering the maximum entropy principle, emergent HOIs, and nonlinear network dynamics as theoretically intertwined notions. As showcased here, such an integrated framework reveals how information encoding, retrieval dynamics, and memory capacity in neural networks are mediated by HOIs, providing principled, analytically tractable tools and insights from statistical mechanics and nonlinear dynamics. More generally, the framework presented in this work extends beyond neural networks and contributes to a general theory of HOIs, paving the road toward a principled study of higher-order phenomena in complex networks.

## Data availability

The CIFAR-100 dataset used in this study is available at https://www.cs.toronto.edu/~kriz/cifar.html.

## Code availability

The code generated in this study is available in the GitHub repository, https://github.com/MiguelAguilera/explosive-neural-networks.

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

## Acknowledgements

The authors thank Ulises Rodriguez Dominguez for valuable discussions on this manuscript. M.A. is funded by a Junior Leader fellowship from 'la Caixa' Foundation (ID 100010434, code LCF/BQ/PI23/11970024), John Templeton Foundation (grant 62828), Basque Government ELKARTEK funding (code KK-2023/00085) and Grant PID2023-146869NA-I00 funded by MICIU/AEI/10.13039/501100011033 and cofunded by the European Union, and supported by the Basque Government through the BERC 2022-2025 program and by the Spanish State Research Agency through BCAM Severo Ochoa excellence accreditation CEX2021-01142-S funded by MICIU/AEI/10.13039/501100011033. P.A.M. acknowledges support by JSPS KAKENHI Grant Number 23K16855, 24K21518. F.R. is supported by the UK ARIA Safeguarded AI programme and the PIBBSS Affiliatership programme. H.S. is supported by JSPS KAKENHI Grant Number JP 20K11709, 21H05246, 24K21518, 25K03085.

## Author contributions

M.A., P.A.M., F.E.R., and H.S. designed and reviewed the research and wrote the paper. M.A. contributed the analytical and numerical results. P.A.M. contributed part of the analytical results of the replica analysis.

## Competing interests

The authors declare no competing interests.
