## [Transparent Peer Review file · Nature Communications]

Explosive neural networks via higher-order interactions in curved statistical manifolds

Corresponding Author: Dr Miguel Aguilera

Version 0:

Reviewer comments:

Reviewer #1

(Remarks to the Author)

In their work, entitled “Explosive neural networks via higher-order interactions in curved statistical manifolds”, the authors focus on mean-field higher-order neural networks and examine how these models can be derived by a maximum-entropy-principle (MEP) exhibiting a relatively small number of parameters. They start from the Shannon entropy encoding for low-order interactions and introduce a deformation parameter which “curves” the underlying statistical manifold in such a way that the effective model contains interactions of all orders, but without a proliferation of the parameters.

I find that the topic considered in this work is interesting and displays the potential of attracting a wide readership. However, in my opinion, there are some issues that the authors should consider before I can recommend publication.

1) In the abstract, the authors mention a “lack of tractable standard models.”, while, in fact, in the last few years many high-order models have been successfully treated, even at a rigorous level. For instance, beyond Ref. [21], I would mention the works by Bovier and Niederhasuer [“The Spin-Glass phase-transition in the Hopfield model with p-spin interactions” — Adv. Theor. Math. Phys. '01] and by Agliari et al. [“Generalized Guerra’s interpolation schemes for dense associative neural networks” — Neur. Netws. '20; “Non-linear PDEs approach to statistical mechanics of dense associative memories” — J. Math. Phys. '22]

2) Still in the abstract, the authors refer to their model as “parsimonious”: please specify with respect to what kind of resource — if I understood correctly this is meant in terms of number parameters.

3) At the end of pag. 1, the authors vaguely state that their framework can contribute to the “explorations of various aspects of HOIs using techniques including mean-field approximations, quenched disordered analysis...” I recommend that the authors rephrase this sentence trying to be more precise on the kind of improvement this framework can lead to because, as recalled in point 1, several features of HOI model are well-known.

4) By reading the introduction, I could not really grasp the novelties introduced in this work. Please sharpen the message and the main results of the work.

5) After Eq. 4, the authors state that “the deformed manifold contains interactions of all orders”, but, as far as I understand, the number of parameters is finite, therefore, I expect that the magnitudes of high-order interactions display some correlation and cannot be tuned independently. Is this the case? A more extensive discussion is in order here to emphasise which kind of constraints are physically implemented by this method. As far as I know, with standard MEP, we ask that, for instance, the expectation, the variance, etc. of some degree of freedom coincide with those found experimentally. Which kind of constraints are set here?

6) At the beginning of Sec. IIB, I would rather refer to the Mattis model [“Solvable spin systems with random interactions” — Phys. Lett. '76]

7) In Sec. III, the authors refer only to the replica trick as a method by which dense associative models can be address to determine the memory capacity, but, as stressed in point 1 this is only one out of other methods (see for instance the works by Bovier, Gayrard, Picco, Shcherbina just to name a few)

8) At the end of pag. 6, the authors describe how the dataset is built up. As far as I understood, they select from CIFAR only those patterns whose covariance is smaller than $10/\sqrt{N}$. Please explain more accurately the underlying procedure, provide evidence and limitations, possibly in the supplementary material.

9) Looking at the phase diagram in Fig. 4, I understand that, for $\gamma=0$ and $T=0.5$, retrieval is feasible as long as $\alpha < 0.06$, recovering, as expected, the standard Hopfield model. Then, by decreasing γ , the capacity increases linearly. On the other hand, when considering the dense Hopfield model or even the exponential Hopfield model the improvement in the capacity is qualitatively more significant. Other versions of the pairwise model reach the theoretical upper bound $\alpha=1$, with relatively small efforts; see for instance Kohonen's projector ["Correlation Matrix Memories" — IEEE Trans. Comp. '72]. Please, comment on the computational efficiency of your model.

Reviewer #2

(Remarks to the Author)

The current work, "Explosive Neural Networks via Higher-Order Interactions in Curved Statistical Manifolds" by Aguilera, Morales, Rosas, and Shimazaki, integrates maximum entropy approaches with Rényi entropy formalism to establish a robust framework for enhancing neural network capacity in storing and retrieving complex patterns. This integration within curved statistical manifolds not only allows for a richer representation of the underlying data structures but also aids in maintaining robustness and diversity in neural network behaviors.

This theoretical framework is particularly promising for studying higher-order phenomena in complex networks, potentially encouraging future insights into non-linear dynamics and emergent behaviors typical in neural systems. Given its innovative approach, I believe the manuscript holds significant potential for publication following revisions.

However, several minor concerns need addressing to enhance the manuscript's clarity and scientific rigor:

1. Implementation of Self-Regulating Annealing Process: A detailed explanation of how the self-regulating annealing process is implemented is crucial. This process can optimize the network's ability to reach an equilibrium and nonequilibrium state, which is essential for investigating higher-order phenomena in complex networks. To broaden the work's impact, it would be beneficial to incorporate experimental neurophysiology datasets that demonstrate higher-order correlations within neuronal populations. A comparative analysis using classical associative memories alongside the self-regulating annealing process with a neurophysiology data set (depicting higher order correlations) could highlight the advantages of this method in accelerating memory retrieval, thus appealing to a wider scientific audience.
2. Enhanced Memory Capacity of Deformed Associative Memory Networks: The manuscript discusses deformed associative memory networks and their enhanced memory capacity, which is a significant advancement. Notably, machine learning techniques have greatly improved associative memory networks, enabling modern architectures like dense associative memories and Hopfield networks to store and retrieve a high volume of memories reliably. Furthermore, recurrent neural networks can optimize memory update rules through advanced training algorithms, such as gradient descent. A discussion on how the current methodology differs from these established approaches would strengthen the manuscript. Exploring specific advantages, such as computational efficiency or improved generalization in the presence of noise, could further elucidate the contribution of this work to the field.
3. Inclusion of Recent References: I recommend incorporating recent studies that delve into the role of the Rényi deformation parameter in exploring higher-order correlations:

-Guisande N and Montani F (2024) Rényi entropy-complexity causality space: a novel neurocomputational tool for detecting scale-free features in EEG/EEG data. *Front. Comput. Neurosci.* 18:1342985. doi: 10.3389/fncom.2024.1342985

-Jauregui, M., Zunino, L., Lenzi, E., Mendes, R., and Ribeiro, H. (2018). Characterization of time series via rényi complexity —entropy curves. *Phys. A* 498, 74–85. doi: 10.1016/j.physa.2018.01.026

Referencing these studies would provide a broader context for the current work's significance and applications.

By addressing these concerns, the manuscript can significantly enhance its clarity and impact, thus making a compelling case for publication.

Reviewer #3

(Remarks to the Author)

Summary:

The paper introduces an analytically tractable associative memory network obtained using the deformed exponential family framework. When the deformation parameter γ is equal to zero, the network reduces to a classical Hopfield network with binary spins and purely pairwise associations. However, different values of γ lead to models with higher order associations, which can be identified through a Taylor expansion of the log-probability. The resulting Glauber dynamics and mean-field theory are analogous to those of a Hopfield model with an adaptive energy-dependent effective temperature. The authors show that the deformed model is as tractable as the original Hopfield networks while showing interesting new thermodynamic phenomena such as discontinuous bifurcations of the order parameters.

General Assessment:

I find the paper to be very interesting as it presents and thoroughly analyzes a new tractable associative memory model with an interesting phenomenology and with interesting links with existing machine learning methods. In particular,

the idea of a state dependent temperature could lead to interesting connections with generative diffusion models and offer guidance on their "noise-scheduling" functions, potentially offering an adaptive approach. I am really pleased by the simplicity of the formulas, which are as straightforward and insightful as those obtained in classical Hopfield networks and Ising models.

I do not have the expertise to check the details and intricacies of the replica calculations and I refer the judgment on their correctness to the other reviewers.

Considerations and questions:

1) While it is obvious that the model exhibits higher-order interactions, their nature remains rather opaque and it is much more subtle than the kind of interactions considered in Dense Associative Memories. The analysis shows that the effect of the higher-order interactions can be reduced to an effective temperature parameter which gives rise to possible positive-feedback loops in the Glauber dynamics.

Therefore, it seems to me that these interactions are not expressive, in the sense that they cannot properly 'fit' the higher order moments of the data. Am I correct?

2) Can you justify the use of the standard Hebb learning rule (Eq. 7) when γ differs from 1? In general, it would be useful to provide a more extensive treatment of the problem of learning, even if you cannot derive a new learning rule that exploits the curved geometry. Could we derive an optimal learning rule for a given value of γ ?

3) In classical Hopfield and other energy-based models, the energy is only relevant up to a constant since the dynamics is determined by the gradient alone. This is not true in this curved model as the absolute energy defines the scale of the effective temperature. Can you provide an intuitive explanation for this behavior?

4) Can the adaptive temperature be related to the noise schedule in generative diffusion models? There seem to be some tantalizing analogies since in generative diffusion the energy landscape becomes sharper as the particle approaches the noise-free training points.

Version 1:

Reviewer comments:

Reviewer #1

(Remarks to the Author)

The authors have satisfactorily addressed all the points raised in my previous report and I can now recommend publication in NComms.

Reviewer #2

(Remarks to the Author)

After carefully reviewing the updated version, I am pleased to confirm that the authors have addressed all my concerns and suggestions thoroughly. The revisions made to the manuscript are comprehensive and significantly improve the clarity, accuracy, and overall quality of the work. The additional explanations, data, and modifications to the text have effectively resolved the issues raised during the initial review.

I appreciate the authors' efforts in responding to the feedback and am satisfied with the current version of the manuscript. However, I would like to highlight one remaining point: to ensure transparency and reproducibility, it is essential that the code and dataset associated with this study be made publicly available online. Providing access to these materials will greatly enhance the impact and utility of the research for the scientific community.

In my opinion, the paper now meets the standards required for publication, pending the availability of the code and dataset.

Reviewer #3

(Remarks to the Author)

I am very happy with the revision in the manuscript and I am fully satisfied by the extensive replies submitted by the authors.

In particular, I find the analysis of the connection with diffusion processes fascinating, and I recommend the authors to summarize it in the conclusion as a promising research direction.

All in all, I recommend this paper for acceptance wholeheartedly.

Color code

Reviewers' comments: black

Our responses: blue

Dear Editors and Reviewers,

We would like to thank you for the time and effort invested in reviewing our manuscript, "Explosive neural networks via higher-order interactions in curved statistical manifolds." We appreciate the constructive feedback and suggestions provided by all three reviewers, which significantly help us to improve the quality of our manuscript. We have carefully considered all the points raised and have made the necessary revisions to address them.

Below is our point-by-point response to each reviewer's comments. Additionally, we provide a file that highlights all changes in the revised manuscript, as requested.

REVIEWER COMMENTS**Reviewer #1 (Remarks to the Author):**

In their work, entitled "Explosive neural networks via higher-order interactions in curved statistical manifolds", the authors focus on mean-field higher-order neural networks and examine how these models can be derived by a maximum-entropy-principle (MEP) exhibiting a relatively small number of parameters. They start from the Shannon entropy encoding for low-order interactions and introduce a deformation parameter which "curves" the underlying statistical manifold in such a way that the effective model contains interactions of all orders, but without a proliferation of the parameters.

I find that the topic considered in this work is interesting and displays the potential of attracting a wide readership. However, in my opinion, there are some issues that the authors should consider before I can recommend publication.

We thank the reviewer for the positive feedback and constructive suggestions. Please find below responses point by point to the different issues raised.

1) In the abstract, the authors mention a "lack of tractable standard models.", while, in fact, in the last few years many high-order models have been successfully treated, even at a rigorous level. For instance, beyond Ref. [21], I would mention the works by Bovier and Niederhasuer ["The Spin-Glass phase-transition in the Hopfield model with p-spin interactions" — Adv. Theor. Math. Phys. '01] and by Agliari et al. ["Generalized Guerra's interpolation schemes for dense associative neural networks" — Neur. Netws. '20; "Non-linear PDEs approach to statistical mechanics of dense associative memories" — J. Math. Phys. '22]

We thank the reviewer for raising this issue. We completely agree that significant progress has been made in the analysis of collective interactions in recent years — particularly, p-spin models and Kuramoto models with low-order higher-order interactions (HOIs). Accordingly, we have revised the abstract to clarify our statement, acknowledging the existence of specific tractable models. We have also included references to the works by Bovier and Niederhasuer (Adv. Theor. Math. Phys. '01) and Agliari et al. (Neur. Netws. '20; J. Math. Phys. '22) mentioned by the reviewer in the *Introduction* section of the revised manuscript. It is important to note, however, that these models account for specific types of higher-order interactions — usually either p-th order or low-order HOIs, typically up to the third or fourth order. To further highlight recent advancements in this area, we referred to recent developments in modeling neuronal nonlinearity by Santos (arXiv, '24) and Hoover (arXiv, '24), and also to papers investigating relativistic Hopfield networks (described in the next paragraph). Further, we have changed the phrasing in the abstract to: "the scarcity of tractable models incorporating interactions of diverse orders"

We have also included paragraphs in the *Discussion section of our revised manuscript* discussing other probabilistic neural network models that account for all orders of interactions such as "relativistic Hopfield networks", which use a specific nonlinear function to induce HOIs (Demircigil '17, Barra '18, Agliari '19). In these works, the authors successfully performed rigorous theoretical analyses. We provided a unifying perspective to explain the behavior of this model through the lens of temperature rescaling and the self-regulated annealing process described in this study. The relativistic Hopfield model corresponds to positive deformed parameter γ .

Inspired by these connections, we also examined the phase diagram of our deformed associative memories within the positive γ regime (see new Figs. 4 and 5c). Our findings indicate a reduced memory capacity compared to the classical model, yet this variant exhibits robust memory retrieval by effectively removing spurious memories.

We thank the reviewer again for directing our attention to these prior works, which has substantially deepened our understanding of our own model.

2) Still in the abstract, the authors refer to their model as "parsimonious": please specify with respect to what kind of resource — if I understood correctly this is meant in terms of number parameters.

The reviewer is correct that "parsimonious" refers to the number of parameters in our model. Accordingly, we have revised the abstract and the text to specify this clearly.

3) At the end of pag. 1, the authors vaguely state that their framework can contribute to the "explorations of various aspects of HOIs using techniques including mean-field approximations, quenched disordered analysis..." I recommend that the authors rephrase this sentence trying to be more precise on the kind of improvement this framework can lead to because, as recalled in point 1, several features of HOI model are well-known.

Thanks for the useful suggestion. We agree that there are rigorous theoretical studies on models with HOIs, and that a more precise description of our framework's contributions is needed. Our model's primary innovation lies in capturing interactions across all orders of higher-order interactions (HOIs) with a minimal number of parameters that are directly interpretable from a generalized MaxEnt principle. To emphasize that we added another yet principled instance that allows theoretical analyses, we rephrased the sentence mentioned above as follows:

The framework adds a minimal model to the previously suggested higher-order models that have rich connections with the literature on the statistical physics of neural networks \cite{bovier2001spin,agliari2020generalized,agliari2023dense,demircigil2017model}, enabling explorations of various aspects of HOIs using techniques including mean-field approximations, quenched disorder analyses, and path integrals.

4) By reading the introduction, I could not really grasp the novelties introduced in this work. Please sharpen the message and the main results of the work.

Many thanks for this feedback. We have re-written the last paragraph to better emphasise the main contributions of this work more clearly. It now reads as follows:

Our analyses reveal how relatively simple curved neural networks exhibit some of the hallmark characteristics of higher-order phenomena, such as explosive phase transitions, arising both in mean-field models and in more complex transitions to spin-glass states. These phenomena are driven by a self-regulated annealing process, which accelerates memory retrieval through positive feedback between energy and an `effective' temperature. Furthermore, we show --- both analytically and experimentally --- that this mechanism can lead to an increase in the memory capacity or robustness of memory retrieval in these neural networks. Overall, the core contributions of this work are (i) the development of a parsimonious neural network model based on the maximum entropy principle that captures interactions of all orders, (ii) the discovery of a self-regulated annealing mechanism that can drive explosive phase transitions, and (iii) the demonstration of enhanced memory capacity resulting from this mechanism.

5) After Eq. 4, the authors state that "the deformed manifold contains interactions of all orders", but, as far as I understand, the number of parameters is finite, therefore, I expect that the magnitudes of high-order interactions display some correlation and cannot be tuned independently. Is this the case? A more extensive discussion is in order here to emphasise which kind of constraints are physically implemented by this method. As far as I know, with standard MEP, we ask that, for instance, the expectation, the variance, etc. of some degree of freedom coincide with those found experimentally. Which kind of constraints are set here?

Thanks for raising these interesting points. Let us first address the latter observation. The deformed model is obtained by maximizing the Rényi entropy under the constraint of the expectation of the features (Eqs. 1, 2), similar to the standard maximum Shannon entropy

modeling. In the revised manuscript, we clarified this point by adding the following sentence before deriving the model under the maximum Rényi entropy principle:

"while constraining $\langle f_a(\mathbf{x}) \rangle$ (i.e., the expectation of features by $p(\mathbf{x})$)"

Please see Supplementary Note 1 for the derivation of the deformed exponential family with this constraint.

Regarding the first point, the reviewer is correct that we parameterize the HOIs using a single deformation parameter, γ ; therefore, the HOIs are inherently related or structured. The necessity of all HOIs and their induced structure can be observed in Eq. 4, which provides the Taylor expansion representation of the model within the framework of the regular exponential family distribution. This expression shows that the deformed manifold contains interactions of all orders even if $f_a(\mathbf{x})$ is restricted to low orders, thereby avoiding a combinatorial explosion of the number of required parameters. To emphasize that this model contains specific structured HOIs, we have added the following statement to the manuscript:

", while giving specific dependency structure across the orders"

To add further clarity, we have provided a more detailed discussion of the structure of the HOIs resulting from the deformed exponential family in the *Discussion* section of the revised manuscript.

6) At the beginning of Sec. IIB, I would rather refer to the Mattis model ["Solvable spin systems with random interactions" — Phys. Lett. '76]

We thank the reviewer for bringing this up. Our revised manuscript includes this reference.

7) In Sec. III, the authors refer only to the replica trick as a method by which dense associative models can be address to determine the memory capacity, but, as stressed in point 1 this is only one out of other methods (see for instance the works by Bovier, Gayraud, Picco, Shcherbina just to name a few]

We appreciate the suggestion. We have included references to other methods for analytically solving associative memories in Sec. III, including the ones suggested by the reviewer.

See also our response to 3).

8) At the end of pag. 6, the authors describe how the dataset is built up. As far as I understood, they select from CIFAR only those patterns whose covariance is smaller than $10/\sqrt{N}$. Please explain more accurately the underlying procedure, provide evidence and limitations, possibly in the supplementary material.

We thank the reviewer for this suggestion, which helps us to clarify relevant details of this experiment. We have included an explanation about the significance of correlations in

memory patterns and the procedure used to mitigate them (note that the procedure is a simple random search) in the main text of the revised manuscript. The new text reads as follows:

Note that associative memories (as well as our theory above) usually assume that patterns are relatively uncorrelated, and specific methods are required to adapt them to correlated patterns \cite{fontanari1990storage,agliari2013parallel}. To simplify the problem, we conducted experiments using a selection of 100 images with covariance values smaller than $10\sqrt{N}$ (the standard deviation of covariance values for uncorrelated patterns is $1\sqrt{N}$). We used a random search to select patterns with low correlations: we randomly picked an image and replaced it if its correlation exceeded the threshold, repeating until all correlations were below it.

9) Looking at the phase diagram in Fig. 4, I understand that, for $\gamma'=0$ and $T=0.5$, retrieval is feasible as long as $\alpha < 0.06$, recovering, as expected, the standard Hopfield model. Then, by decreasing γ' , the capacity increases linearly. On the other hand, when considering the dense Hopfield model or even the exponential Hopfield model the improvement in the capacity is qualitatively more significant. Other versions of the pairwise model reach the theoretical upper bound $\alpha=1$, with relatively small efforts; see for instance Kohonen's projector ["Correlation Matrix Memories" — IEEE Trans. Comp. '72]. Please, comment on the computational efficiency of your model.

The reviewer raises a fair point. Indeed, various methods (such as weight adjustment based on correlations or models like Dense Associative Memories) have been developed to increase memory capacity of these systems, achieving levels where the storage capacity parameter α approaches or even exceeds 1. These models have been specifically engineered to minimize basin overlap in state space, either by creating greater separation between basins or by narrowing the basins themselves.

That being said, please note that the objective in our investigation is not to maximize memory capacity—a challenge already addressed by such engineered models—but, rather, to develop a parsimonious model that explains high-order phenomena in neural systems and its relationship with explosive phase transitions, within a principled yet tractable framework.

Interestingly, we found that positive γ of this model reduces the area of the mixed phase, where spurious memories emerge, therefore making memory retrieval more robust (See new Fig.4). Furthermore, we note that many of the modifications traditionally used to enhance memory capacity could be adapted to our curved MaxRenyiEnt model, potentially extending the memory capacity even further. Exploring the potential intersection between these methods is an interesting direction for future work.

To make this important point clear, we have added a clarification of these issues and the references mentioned by the in the discussion (third paragraph from the end).

Reviewer #2 (Remarks to the Author):

The current work, "Explosive Neural Networks via Higher-Order Interactions in Curved Statistical Manifolds" by Aguilera, Morales, Rosas, and Shimazaki, integrates maximum entropy approaches with Rényi entropy formalism to establish a robust framework for enhancing neural network capacity in storing and retrieving complex patterns. This integration within curved statistical manifolds not only allows for a richer representation of the underlying data structures but also aids in maintaining robustness and diversity in neural network behaviors.

This theoretical framework is particularly promising for studying higher-order phenomena in complex networks, potentially encouraging future insights into non-linear dynamics and emergent behaviors typical in neural systems. Given its innovative approach, I believe the manuscript holds significant potential for publication following revisions.

We appreciate the reviewer's positive evaluation of our work, and the recognition of our framework's potential to advance the study of higher-order phenomena and complex systems.

However, several minor concerns need addressing to enhance the manuscript's clarity and scientific rigor:

1. Implementation of Self-Regulating Annealing Process: A detailed explanation of how the self-regulating annealing process is implemented is crucial. This process can optimize the network's ability to reach an equilibrium and nonequilibrium state, which is essential for investigating higher-order phenomena in complex networks. To broaden the work's impact, it would be beneficial to incorporate experimental neurophysiology datasets that demonstrate higher-order correlations within neuronal populations. A comparative analysis using classical associative memories alongside the self-regulating annealing process with a neurophysiology data set (depicting higher order correlations) could highlight the advantages of this method in accelerating memory retrieval, thus appealing to a wider scientific audience.

We thank the reviewer for their fair assessment and insightful suggestions. We fully agree that the self-regulating annealing process is one of the key findings in this study, and considering its implications in neural populations is an important avenue for future research. Following this suggestion, we rewrote substantial portions of the *Discussion* section to clarify implications of the scaling of the inverse temperature and self-regulated annealing process, and added a paragraph to relate our findings with neuronal HOIs.

Along these lines, let us note that previous work has already provided substantial evidence demonstrating the relevance of higher-order correlations within neuronal populations. For example, previous work on neural higher-order interactions (HOIs) have identified an alternating pattern of HOIs, characterised by positive pairwise and negative third-order interactions. This structure has been recognized as a signature of sparse population activity. Indeed, such an alternating HOI structure can be constructed within the deformed exponential family, for example, by setting a negative bias H with a positive γ , although other configurations are also possible. We thus strongly agree with the reviewer's

suggestion that identifying configurations that capture neural HOIs effectively is a crucial research direction.

We also agree with the reviewer that doing comparative analyses of classical associative memories alongside self-regulating annealing on neurophysiological data would be extremely interesting. However, it is important to note that this is highly non-trivial, requiring technical developments that we believe are better suited for an extension of this manuscript. Specifically, besides the need for data collection and estimation methods, performing the 'inverse Ising problem' in this framework (i.e., the inference of the parameters of the deformed networks from the data) would need new techniques. Indeed, the required computation for learning the corresponding parameters is non-trivial as the dual-coordinates of the deformed exponential family distributions is not a simple Legendre transform of the natural parameters. As a matter of fact, we are currently working on the mean-field method to estimate the expectation parameters of the deformed exponential family distributions, which is a prerequisite for the parameter estimation in the large-scale data.

We believe that addressing these points with rigorous methodologies represents a significant effort which would deserve a separate publication. Accordingly, we have added a remark in the *Discussion* section of the revised manuscript stating that the inverse problem for the deformed neural networks is an important — yet still untouched — research area, and hence that advanced statistical methods are required to accurately assess the effectiveness of the proposed model in modelling the real neuronal dynamics.

2. **Enhanced Memory Capacity of Deformed Associative Memory Networks:** The manuscript discusses deformed associative memory networks and their enhanced memory capacity, which is a significant advancement. Notably, machine learning techniques have greatly improved associative memory networks, enabling modern architectures like dense associative memories and Hopfield networks to store and retrieve a high volume of memories reliably. Furthermore, recurrent neural networks can optimize memory update rules through advanced training algorithms, such as gradient descent. A discussion on how the current methodology differs from these established approaches would strengthen the manuscript. Exploring specific advantages, such as computational efficiency or improved generalization in the presence of noise, could further elucidate the contribution of this work to the field.

Thank you for this valuable suggestion. While our primary goal was to construct a parsimonious model of higher-order interactions (HOIs) from first principles, we agree that it is essential to clarify how our approach differs from established methods and to highlight the advantages of our framework.

In considering how our approach differs from others, we concluded that the concepts of temperature rescaling and the self-regulated annealing process are broadly applicable to networks that deform the energy landscape using a nonlinear function. We thank the reviewer for raising this point. To address it, we substantially modified the *Discussion* section and explained how other networks with HOIs can be interpreted through the lens of temperature rescaling.

Further, to address the specific advantages raised by the reviewer, we expanded our parameter exploration to include positive gamma values (see new Fig. 4). Notably, our results indicate that a positive gamma shrinks the mixed phase—where spurious memories emerge—thus enhancing the system’s robustness for memory recall. This finding highlights one of the key advantages of our modeling framework.

We sincerely thank the reviewer for prompting us to reflect on these important and unique contributions.

3. Inclusion of Recent References: I recommend incorporating recent studies that delve into the role of the Rényi deformation parameter in exploring higher-order correlations:

-Guisande N and Montani F (2024) Rényi entropy-complexity causality space: a novel neurocomputational tool for detecting scale-free features in EEG/iEEG data. *Front. Comput. Neurosci.* 18:1342985. doi: 10.3389/fncom.2024.1342985

-Jauregui, M., Zunino, L., Lenzi, E., Mendes, R., and Ribeiro, H. (2018). Characterization of time series via rényi complexity—entropy curves. *Phys. A* 498, 74–85. doi: 10.1016/j.physa.2018.01.026

Referencing these studies would provide a broader context for the current work’s significance and applications.

We thank the reviewer for this suggestion, which helps us to contextualise our contribution better. Indeed, these references are good examples that show the distributions derived under the Rényi entropy are suitable to model natural signals. Accordingly, our revised manuscript includes these references to motivate the use of the Rényi entropy.

By addressing these concerns, the manuscript can significantly enhance its clarity and impact, thus making a compelling case for publication.

We thank the reviewer again for the thoughtful suggestions and the constructive feedback.

Reviewer #3 (Remarks to the Author):

Summary:

The paper introduces an analytically tractable associative memory network obtained using the deformed exponential family framework. When the deformation parameter gamma is equal to zero, the network reduces to a classical Hopfield network with binary spins and purely pairwise associations. However, different values of gamma lead to models with higher order associations, which can be identified through a Taylor expansion of the log-probability. The resulting Glauber dynamics and mean-field theory are analogous to those of a Hopfield model with an adaptive energy-dependent effective temperature. The authors show that the

deformed model is as tractable as the original Hopfield networks while showing interesting new thermodynamic phenomena such as discontinuous bifurcations of the order parameters.

General Assessment:

I find the paper to be very interesting as it presents and thoroughly analyzes a new tractable associative memory model with an interesting phenomenology and with interesting links with existing machine learning methods. In particular, the idea of a state dependent temperature could lead to interesting connections with generative diffusion models and offer guidance on their "noise-scheduling" functions, potentially offering an adaptive approach. I am really pleased by the simplicity of the formulas, which are as straightforward and insightful as those obtained in classical Hopfield networks and Ising models.

I do not have the expertise to check the details and intricacies of the replica calculations and I refer the judgment on their correctness to the other reviewers.

We thank the reviewer for the positive comments on our work.

Considerations and questions:

1) While it is obvious that the model exhibits higher-order interactions, their nature remains rather opaque and it is much more subtle than the kind of interactions considered in Dense Associative Memories. The analysis shows that the effect of the higher-order interactions can be reduced to an effective temperature parameter which gives rise to possible positive-feedback loops in the Glauber dynamics.

Therefore, it seems to me that these interactions are not expressive, in the sense that they cannot properly 'fit' the higher order moments of the data. Am I correct?

We thank the reviewer for raising this issue, which helps us to clarify the contribution of our work. The reviewer is correct in highlighting that our proposed model is not capable of perfectly fitting any arbitrary joint probability distribution. In contrast, our model uses the maximum entropy principle to provide a model with a relatively small number of degrees of freedom which nonetheless span a rich taxonomy of high-order behaviours.

Moreover, our results show that adding a single additional degree of freedom is enough to elicit explosive phase transitions and an expansion on the memory capacity. This new parameter (γ) includes several different high-order interactions, these in specific mixtures depending on its value (see Equation 4). In our mean-field solution for associative memories, this is translated into an effective temperature.

To make this clearer, we have revised the text as follows. In the revised manuscript, we clarified the constraints used under the maximum Rényi entropy principle.

while constraining $\langle f_a(\mathbf{x}) \rangle$ (i.e., the expectation of features by $p(\mathbf{x})$)

Further, to emphasize that this model contains specific structured HOIs, we added the following statement near Eq.4:

, while giving specific dependency structure across the orders

We provided a more detailed discussion of the structured HOIs of the deformed exponential family in the *Discussion* section of the revised manuscript.

Please also see our response to Reviewer #1 Question 5.

2) Can you justify the use of the standard Hebb learning rule (Eq. 7) when γ differs from 1? In general, it would be useful to provide a more extensive treatment of the problem of learning, even if you cannot derive a new learning rule that exploits the curved geometry. Could we derive an optimal learning rule for a given value of γ ?

We thank the reviewer for this interesting suggestion. Please note that associative memories establish parameters in a way that creates energy minima at specific points (patterns ξ in the manuscript). While this is certainly inspired by Hebb's rule, it certainly does not involve a proper learning process. Hebbian-like learning here is represented by a one-shot process, $J_{ij} = \sum_a \xi_i^a \xi_j^a$, in which we assume that the network has been exposed to a series patterns ξ and modified its couplings according to its correlations. This heuristic approach is effective for any value of γ , making it suitable for the theoretical analysis presented in our paper.

That being said, the reviewer is correct in pointing out that the actual learning process is not explicitly addressed in our analyses. We fully agree that it would be very interesting to address this aspect, but we would also want to acknowledge that properly developing this line of research — by e.g. designing a new learning rule that extends Hebbian learning — would require a substantial amount of theoretical and computational work. Specifically, the required computation for learning the corresponding parameters is non-trivial as the dual-coordinates of the deformed exponential family distributions is not a simple Legendre transform of the natural parameters. We are currently working on the mean-field method to estimate the expectation parameters of the deformed exponential family distributions, which is a prerequisite for the parameter estimation in the large-scale data. Hence, while we are currently working on this subject, we believe it would be ideally suited for a future extension of this work. Motivated by this observation, the *Discussion section in our revised manuscript highlights that* investigating learning methods for curved geometry is an extremely interesting avenue for future work.

3) In classical Hopfield and other energy-based models, the energy is only relevant up to a constant since the dynamics is determined by the gradient alone. This is not true in this curved model as the absolute energy defines the scale of the effective temperature. Can you provide an intuitive explanation for this behavior?

We thank the reviewer for highlighting this intriguing point. Indeed, for models in the exponential family, energy gradients alone suffice to determine the system's dynamics and statistical properties. However, in the context of deformed exponentials, constant energy shifts can alter the system's behavior. One elegant way to address this situation has been proposed by Tsallis, Mendes, and Plastino (1998), who suggested using the statistics of escort distributions as constraints in the maximum entropy framework. Their work demonstrates that the constant-invariant formulation of the maximum Tsallis/Rényi entropy corresponds to a temperature rescaling of the non-invariant form. In other words, a change of coordinate system allows to obtain an equivalent model that is invariant to energy shifts. These shifts can alternatively be considered as changing the form of the constraints on the

maximum entropy solution, reflecting the role of higher-order interactions encoded in the deformation parameter. This mapping illustrates how the deformation parameter governs changes in the statistical structure imposed by the system's constraints.

Motivated by this observation, our revised manuscript includes the following discussion after we introduce the Energy function of the deformed exponential:

Note that, unlike exponential families, these models do not exhibit energy invariance under constant shifts. However, as demonstrated in Ref.~\cite{tsallis1998role}, deformed exponential models can be related to energy-invariant models by rescaling their temperature, which can be seen as maximizing entropy with respect to escort statistics rather than the original natural statistics.

4) Can the adaptive temperature be related to the noise schedule in generative diffusion models? There seem to be some tantalizing analogies since in generative diffusion the energy landscape becomes sharper as the particle approaches the noise-free training points.

We thank the reviewer for this extremely interesting remark. There are several variants in noise scheduling in the diffusion model such as Denoising Diffusion Probabilistic Models. These include linear or quadratic scheduling or cosine scheduling. While these are motivated to realize an annealing process (high to low temperature) to search for global minima, they are neither objectively obtained nor state (or energy) dependent. Exceptions could be some methods that try to learn the scheduling from the data, but they are time-consuming.

In contrast, our noise scheduling naturally appears from sampling of the deformed exponential family. Thus, the noise scheduling in the current diffusion models itself is not directly related to the self-regulated annealing process found in this study. At the same time, however, this self-regulated annealing is widely applicable to the associative networks with deformed nonlinearity (See the revised Discussion). We also note that the diffusion model can be considered as a (continuous-version of) modern Hopfield network with log sum exp nonlinearity [1,2]. Its energy function is given as

$$\mathfrak{F}[\mathbf{m}] = -\frac{1}{\beta(t)} \log \sum_a e^{\beta(t)m_a}$$

Here, we omitted the quadratic term of x for simplicity. The $\beta(t)$ is a scheduled inverse temperature, which is typically given as $\beta(t)=1/(T-t)\sigma^2$. m_a is the Mattis magnetization:

$$m_a = \frac{1}{N} \sum_i \xi_i^a x_i$$

While formal analysis on the continuous-valued model would be more appropriate, it is relatively easy to see that the self-regulated annealing process occurs in the dynamics of the binary patterns following the above nonlinear energy. Further, when x is binary, we also recover a modern Hopfield network with exponential nonlinearity if we replace log with a linear function [3]. Suppose that we flip the k -th neuron, then we have

$$\tilde{m}_a = m_a - \frac{2\xi_k^a x_k}{N}$$

In the limit of large networks, the energy difference used in the sampling procedure is approximated as

$$\beta(t)\Delta\mathfrak{F}[\mathbf{x}] = \beta(t)(\mathfrak{F}[\tilde{\mathbf{m}}] - \mathfrak{F}[\mathbf{m}]) \approx \sum_a \beta(t) \frac{\partial\mathfrak{F}[\mathbf{m}]}{\partial m_a} \frac{2\xi_k^a x_k}{N} = \sum_a \beta'_a(t) \frac{2\xi_k^a x_k}{N}$$

where

$$\beta'_a(t) = \beta(t) \frac{\partial\mathfrak{F}[\mathbf{m}]}{\partial m_a} = \beta(t) \frac{e^{\beta(t)m_a}}{\sum_a e^{\beta(t)m_a}}$$

Consequently, the flip of the k th neuron in agreement with ξ_k^a is accelerated as the pattern ξ^a is retrieved. Since the retrieval of the pattern ξ^μ indicates reduced magnetization of the patterns orthogonal to ξ^μ , the recall of a specific memory decreases the effective inverse temperature of other memories. This mechanism prevents those other memories from being retrieved, thereby reducing inter-pattern interference and improving retrieval accuracy.

Thus, the retrieval process of the (discrete-version of) diffusion model as well as the modern Hopfield networks can also be viewed from the perspective of a self-regulated annealing process, with multiple effective temperatures assigned to each memorized pattern, in addition to the predefined time-dependent inverse temperature $\beta(t)$. While an extensive investigation of the self-regulated annealing processes across various other networks is beyond the scope of the current manuscript, we agree that it represents an important step toward an intuitive understanding of memory retrieval in modern learning machines. Moreover, it lays the groundwork for enhancing noise scheduling based on the results obtained in this work.

We thank the reviewer for guiding us to consider this important problem.

[1] Ambrogioni, L. (2024). In Search of Dispersed Memories: Generative Diffusion Models Are Associative Memory Networks. *Entropy*, 26(5), 381.

[2] Ambrogioni, L. (2023). The statistical thermodynamics of generative diffusion models. arXiv preprint arXiv:2310.17467.

[3] Demircigil, M., Heusel, J., Löwe, M., Upgang, S., & Vermet, F. (2017). On a model of associative memory with huge storage capacity. *Journal of Statistical Physics*, 168, 288-299.